# Exploring the narrative landscape: The discursive construction of identity for Chinese enterprises in Africa

Wenyu Liu[ID]⊚*, Yu Yang⊚

School of Foreign Languages, Dalian University of Technology, Dalian, China

⊚ These authors contributed equally to this work.
* liuwy@dlut.edu.cn

## Abstract

This study investigates the construction of corporate identities by Chinese enterprises in Africa, employing the Three-dimensional Model of Fairchough in conjunction with the corporate communication strategies outlined by Kim and Rader. Through an extensive analysis of official website content from 200 Chinese companies, this study explores how these corporations utilize communication strategies to project their identities. The findings reveal a strategic use of discourse to align corporate practices with local cultures and business norms, facilitating their expansion and acceptance in African markets. This study contributes to understanding the dynamic and constructed nature of corporate identities in cross-cultural settings and provides insights into the strategic communication practices of Chinese firms in Africa.

## 1. Introduction

In recent years, China has solidified its position as Africa's foremost trade ally and a principal source of investment. Chinese enterprises are active across various industries, including manufacturing, technology, infrastructure, and natural resources extraction. These engagements transcend mere economic transactions, reflecting deep cultural, social, and political interactions between the two regions.

In the African context, the corporate identity of Chinese enterprises is crucial for their access and success, often reflecting a convergence of national strategies and cultural values that aim to foster partnerships and mutual growth. The communication strategies of these enterprises have not only facilitated their expansion in Africa but have also played a significant role in shaping the narrative of China-Africa relations. As these entities deepen their involvement with African communities, businesses, and governments, their evolving corporate identities will continue to influence their global business trajectories and impact on shared development agendas [1].

The interplay between identity and discourse is particularly prominent in how organizations articulate their corporate identities through language, narratives, and communication strategies. Although there is extensive research on the corporate identity of multinational corporations in global markets, the distinctive approaches Chinese enterprises employ to discursively construct and navigate their corporate identities in the African landscape remains

**Data availability statement:** All relevant data are within the paper and its Supporting Information files.

**Funding:** The author(s) received no specific funding for this work.

**Competing interests:** The authors have declared that no competing interests exist.

unclear. This study addresses a gap in understanding by investigating how Chinese enterprises operating in Africa construct their corporate identities. The investigation draws upon the Three-dimensional Model of Critical Discourse Analysis (CDA) developed by Norman Fairclough [2] and the Corporate Communication Strategy Typology developed by Kim and Rader [3]. This research will delve into three key questions:

(1)  What corporate identities do Chinese companies in Africa project in their self-narratives?

(2)  What communication strategies do these companies prioritize to shape their identity?

(3)  How do sociocultural dynamics impact the identity formation of Chinese enterprises in Africa?

## 2.  Corporate identity

The concept of Corporate Identity encompasses multiple interpretations, tracing back to its origins in the 1950s, initially introduced by Lippincott and Margulies [4]. Originating in the marketing realm, it initially served commercial interests, focusing on advertisement campaigns and logo design as part of broader business activities. Increasingly, corporate identity is viewed as a strategic asset, pivotal in securing and maintaining a competitive advantage. It represents a process of self-representation, positioning, and differentiation, allowing corporations to establish credibility among various stakeholders and secure an advantageous position in the dynamic business environment [5]. Corporate identity encompasses the structure, roles, and values of organizations, emphasizing continuity, uniqueness, and centrality of the organization [6,7].

Corporate identity is inherently multidimensional and interdisciplinary. It encompasses visual components (e.g., logos and branding) and organizational culture and stakeholder relationships, making it a complex construct [8]. Its interdisciplinary nature is evident in its connections to fields like management, marketing, sociology, psychology, and communication. Furthermore, corporate identity is dynamic and evolving, shaped by factors like corporate strategy, culture, and stakeholder expectations [9], requiring an interdisciplinary approach that integrates insights from marketing, organizational behavior, and communication studies.

Research on corporate identity covers a broad spectrum, including its evolution, components, models, frameworks, and intersections with Corporate Social Responsibility (CSR) and national identity. Corporate identity has been studied from various disciplinary perspectives, each offering unique insights. Economists examine its connection to brand equity, market positioning, and consumer behavior, demonstrating how a strong corporate identity can enhance financial performance and competitive advantage [10–12]. International relations scholars analyze how multinational corporations construct identities that resonate across diverse cultural and regulatory environments, influencing international perceptions and geopolitical dynamics [13–15]. Journalism studies focus on how corporate identity is communicated through media channels, examining how organizational narratives and public relations strategies shape public perception and media representation [16,17].

While current empirical research using discourse analysis has made valuable contributions, much of it focuses on specific industries and compares the strategies of individual companies, often neglecting broader, cross-industry evaluations [18–22]. Additionally, although previous studies have explored corporate identity through mediums such as annual reports and corporate social responsibility statements, there is a notable gap in research examining corporate profiles presented on official websites. These profiles play a critical role in directly shaping corporate identity and offering a comprehensive portrayal of companies, making them a fertile area for further investigation [5,23]. Expanding the scope of discourse analysis to include

such online representations could enrich our understanding of how corporate identity is constructed across various platforms.

## 3. Three-dimensional model of critical discourse analysis

Critical discourse analysis (CDA) serves as a cornerstone in identity research, offering a robust framework that integrates language analysis with social contexts. Afzaal underscores that CDA provides a comprehensive tool for dissecting complex geopolitical narratives [24]. This approach goes beyond merely analyzing spoken or written words, aiming to reveal underlying intentions within social interactions [25–27]. CDA not only examines representations of the real world but also explores how discourse shapes and is shaped by social bonds and identities, thereby illuminating the intricate relationship between discourse and identity [28]. This method offers a sophisticated theoretical base that skillfully combines textual analysis with broader social contexts, exposing the socially constructed nature of discourse.

The Three-dimensional Model proposed by Fairclough is a pivotal component of CDA, consisting of three interconnected stages: the linguistic description of texts, the interpretation of discursive practices, and the explanation of their relationships to social processes [2]. The model initiates with a thorough analysis of the linguistic features of texts. It then examines how these texts act as manifestations of discursive practices in the interpretation phase, and ultimately, it considers the social factors influencing these practices in the explanation stage. For example, Afzaal et al. utilized the model of Fairclough to analyze how the media in Pakistan constructs discourses around the China-Pakistan Economic Corridor (CPEC), revealing that discursive practices shape public perceptions and geopolitical narratives [29]. By addressing both the content and the processes of production and reception, the Three-dimensional Model of Fairclough [2] elucidates the dialectical relationship between language and society, representing a significant evolution in the field of discourse analysis [27].

## 4. Kim and Rader's corporate communication strategy typology

Corporate communication strategy is integral to the overall business strategy of a company, focusing on the strategic dissemination of information to various stakeholders. This involves crafting messages aligned with the company's mission and values, selecting effective communication channels, preparing for potential crises, and ensuring brand consistency. The study of this strategy examines how these elements work together to cultivate a positive corporate identity and maintain robust relationships with both the public and internal staff [30].

Kim and Rader proposed a typology of corporate communication strategies that categorizes approaches into Corporate Ability (CAb) Strategy, Corporate Social Responsibility (CSR) Strategy, and Hybrid Strategy [3]. As demonstrated by Table 1, the CAb Strategy emphasizes showcasing the products and business capabilities of companies to enhance public perception of its strength, while the CSR Strategy focuses on the commitment of companies to social responsibilities like employee care and environmental protection to build trust. The Hybrid Strategy combines these approaches, presenting a comprehensive image that reflects both business prowess and social commitments. This framework underscores the importance of proactive, transparent communication and integrated marketing to maintain consistent messaging across all platforms and prepare for potential communication challenges.

Researchers have utilized corporate communication strategy typology proposed by Kim and Rader in diverse ways. For instance, it has been applied to assess the efficacy of integrated marketing communications in ensuring consistent messaging and preventing communication

**Table 1. Indicators of CAb strategy and CSR strategy.**

| Strategy | Corporate Ability (CAb) Strategy | Corporate Social Responsibility (CSR) Strategy |
|---|---|---|
| Indicators | Expertise in product or service quality | Environmental stewardship |
| | Global success | Philanthropic contribution |
| | Implementation of quality control program | Educational commitments |
| | Industry leadership | Employee involvement |
| | Market orientation | Public health commitments |
| | Innovation and R&D efforts | Sponsorship of cultural activities |

crises [16,17]. Yuan and Sheng conducted a comparative analysis of communication strategies employed by Chinese and Japanese companies on social media platforms [31], such as Facebook and Twitter, focusing on how these strategies influenced public responses and managed customer complaints. In another study, Kim, Park, and Kim explored the effectiveness of communication strategies emphasizing Corporate Advocacy (CA) versus Corporate Social Responsibility (CSR) across different industries, including durable goods and those with negative societal perceptions [32]. Chen and Tao utilized the typology to explore the effectiveness of a hybrid communication strategy combining CA and CSR elements [33], assessing whether this blended approach enhanced recall of CSR information and improved company evaluations compared to a CSR-focused strategy.

The typology is a valuable tool for analyzing corporate communication strategies across industries. It helps researchers assess the alignment of company messaging and responses to communication challenges. The widespread use of this typology in evaluating various communication approaches demonstrates its effectiveness in enhancing our understanding of corporate communications.

## 5. Methodology

### 5.1 Data collection

This analysis relies on data carefully collected from the official websites of Chinese companies in Africa by manually copying and pasting information, which ensures both accuracy and relevance. The list of these companies is primarily derived from three key resources: the Reports on Chinese Investment in Africa (2021-2023) by the China-Africa Business Council (CABC), a list of Member Enterprises from the CABC website, and insights from AllAfrica Reports, a platform aggregating African news. A total of 200 Chinese companies are identified, and relevant content sections from their websites, including "About Us," "Corporate Responsibility," and "Culture," are compiled into a corpus of English-language website content for Chinese Enterprises in Africa (CCIA), encompassing 182,780 tokens. This study analyzes 200 texts, each representing a single company. Table 2 provides a detailed breakdown of the number of texts and tokens collected from different types of Chinese corporations operating in Africa.

**Table 2. The number of texts and tokens.**

| Type of Corporate | Number of texts | Tokens of texts |
|---|---|---|
| State-owned Enterprises | 80 | 88,295 |
| Private Enterprises | 120 | 94,485 |
| Total | 200 | 182,780 |

## 5.2 Analytical tools and procedure

To dissect the corpus, we employ two analytical tools: Maxqda 2020 for coding and Wmatrix 5.0 for text analysis.

Maxqda 2020 facilitates the detailed annotation of the corpus based on the twelve indicators of corporate communication strategies, as defined by Kim and Rader [3]. This process involves tagging excerpts that exemplify attributes of CAb or CSR strategies, such as service quality, innovation, philanthropy, and environmental stewardship. Annotations are organized into an Excel spreadsheet to quantify the presence (1) or absence (0) of these strategic components, allowing for the assessment of the focused communication strategy employed by each company and its emphasized components of its focused communication strategy.

The CCIA corpus is investigated using Wmatrix, which compared it against a reference corpus and utilizes the USAS tagging system to assign semantic domains, facilitating thematic analysis. This analysis identifies the top ten semantic domains, leading to a qualitative examination of high-frequency words and collocations that elucidated various corporate identities. These identities are then mapped against the corporate communication strategy components to identify correspondence and detect unique or unexpected identity manifestations. Finally, an examination based on Excel data investigates the scoring patterns of companies adopting different communication strategies to explore the underlying influential factors reflecting diverse identity constructions.

# 6. Results and discussion

## 6.1 Textual analysis

This study has identified the top ten thematic semantic domains within the CCIA corpus, utilizing the analysis of high-frequency words and collocation indices to outline the thematic characteristics that Chinese companies in Africa emphasize as they construct their corporate identities.

As demonstrated in Table 3, the primary semantic domains identified among these Chinese enterprises include Z5 [Grammatical bin], Z99 [Unmatched], Z8 [Pronouns], A1.1.1 [General actions, making etc.], I2.1 [Business: Generally], S8 + [Helping], M7 [Places], Z2 [Geographical names], N1 [Numbers] and S5 + [Groups and affiliation].

In this study, we have identified 377 unique semantic domain types, which collectively occur 74,086 times. Our research specifically examines the top ten of these domains, as

Table 3. Top 10 semantic domains of CCIA corpus.

| Rank | Semantic domains | Frequency | Relative Frequency |
|---|---|---|---|
| 1 | Z5 | 21034 | 28.39 |
| 2 | Z99 | 2981 | 4.02 |
| 3 | Z8 | 2881 | 3.89 |
| 4 | A1.1.1 | 2030 | 2.74 |
| 5 | I2.1 | 1621 | 2.19 |
| 6 | S8 + | 1353 | 1.83 |
| 7 | M7 | 1294 | 1.75 |
| 8 | Z2 | 1284 | 1.73 |
| 9 | N1 | 1280 | 1.73 |
| 10 | S5 + | 1107 | 1.49 |

outlined in Table 3. Table 3 presents the frequency and relative frequency of each semantic domain, illustrating its proportion of the total frequency. Relative frequency is determined as follows:

$$\text{Relative Frequency} = \left(\frac{\text{Frequency of a specific semantic domain}}{\text{Total frequency of all semantic domains}}\right) \times 100$$

For example, semantic domain Z2, with a frequency of 21,034, has a relative frequency of 28.39%, indicating its significant presence within the dataset.

Within the semantic domain Z5, prepositions and conjunctions such as "and," "the," "of," "to," "in" are considered of marginal importance and are thus omitted from the analysis. Similarly, specific high-frequency words like company names (CRBC, Lenovo, Sepco, Xiaomi) categorized under Z99 are regarded outside the research scope. Instead, the study concentrates on eight other semantic domains, conducting a detailed examination and providing examples from each to explore how these Semantic Meaning Categories (SMCs) contribute to shaping the corporate identities of Chinese firms in Africa.

The semantic domains are further organized into six primary categories in Table 4 for analysis. The first category combines A1.1.1 [General actions, making, etc.] with I2.1 [Business: General]; the second includes S8 + [Helping]; the third encompasses Z2 [Geographical names] and M7 [Places]; the fourth is dedicated to N1 [Numbers]; the fifth involves S5 + [Groups and affiliation]; and the final category focuses on Z8 [Pronouns]. This comprehensive analysis of each category aims to critically assess high-frequency words and their collocations, unveiling how these enterprises construct their identities through textual strategies, thereby emphasizing the narrative and representational tactics employed to forge their corporate identities.

**6.1.1 Analysis of category 1.** Table 4 shows that the first category, encompassing A1.1.1 [General actions, making, etc.] and I2.1 [Business: General], defines the core identity of professional service and product providers, distinguishing them from other sectors. This distinction is underscored by frequent terms listed in Table 5, such as "project" and "manufacturing," which emphasize foundational actions and general business practices, highlighting the specialized role these enterprises play within the competitive landscape.

Table 5 illustrates the frequent words and collocations within the semantic domain A1.1.1 in Category 1. The semantic domain A1.1.1 [General actions, making, etc.] serves to portray actions and behaviors of companies, employing a mix of verbs and nouns to depict its operations, initiatives, and projects. Nouns such as "projects" in Example 1 and Example 2, offering a window into the core operations that define the business landscape. The integration of the terms "world-class" and "international" with "projects" illustrates the broadening of China's business activities in Africa to a global scale. The alignment of "electrical," "energy," and "power" with "projects" highlights a concentration on enhancing infrastructure in areas

**Table 4. Categories of 8 semantic domains.**

| Category | Semantic domain |
|---|---|
| 1 | A1.1.1 [General actions, making etc.], I2.1 [Business: Generally] |
| 2 | S8 + [Helping] |
| 3 | Z2 [Geographical names], M7 [Places] |
| 4 | N1 [Numbers] |
| 5 | S5 + [Groups and affiliation] |
| 6 | Z8 [Pronouns] |

**Table 5. High-frequency words and collocations of semantic domain A1.1.1 in Category 1.**

| Semantic domain | High-frequency words | Example | Collocations |
|---|---|---|---|
| A1.1.1 | projects(182),operation(113),production(99),manufacturing(58),activities(42) | projects (182) | world-class,international; social,community; electrical,energy,power |
| | create(77),committed(86), make(63),do(30),implement(20) | create (77) | value,benefits, a better future, opportunities,workplace, |
| | | make (63) | customers(successful), employees(happy), contributions, efforts, |

like electricity and energy. The fusion of "social" and "community" with "projects" underlines a mutual dedication to providing products and services while also giving importance to the advancement of communities in African countries.

**Example 1.** In 2022, CMOC invested a total of RMB 290.4 million (eq. US$43.5 million) in community **projects**.

**Example 2.** The second step is to "ramp up with multiplying production capacity" as the two world-class **projects**.

Meanwhile, the performative verbs "create," "make," "do," and "implement" signify the proactive measures a company undertakes to achieve its objectives, underlining the intent to serve as an active participant in its industry. The frequent use of "make" within the A1.1.1 domain, which encompasses making "customers (successful)," "employees (happy)," along with "contributions" and "efforts," reflects the commitment of companies to the third party and social contribution. Example 3 highlights the emphasis on prioritizing employee and customer well-being and satisfaction, aiming to cultivate lasting relationships, loyalty, and sustained growth.

**Example 3.** Its mission is to **make** customers succeed, **make** employees happy, and contribute to the society.

Similarly, the notion of "making contributions" in Example 4 highlights the company's efforts to tackle social and environmental issues, aiming for a more sustainable and fair world.

**Example 4.** In responsibility practice, HE continuously **make** active contributions to sustained and stable development of the enterprise and to social harmony and stability.

The presence of "create" within the A1.1.1 semantic domain, emphasizing "benefits" in Example 5 and "workplace" in Example 6, highlights the commitment of companies to value creation, delivering benefits to both customers and employees, fostering a promising future, uncovering opportunities, and enhancing convenience for stakeholders and society at large. These initiatives underscore the company's forward-thinking approach and responsible engagement, solidifying its role as a positive contributor to both economic and social development.

**Example 5.** SEPCO will continue to **create** value-added benefits for customers and realize their good wishes.

**Example 6.** We **create** a workplace of equal opportunities, respect and diversity.

Additionally, the utilization of the term "committed" within the A1.1.1, as illustrated in Examples 7 and 8, profoundly echoes the principles of perseverance and unwavering resolve in achieving the company's goals and notable successes. These characteristics depict the enterprises as dedicated and visionary entities, committed to their long-term objectives and demonstrating an enduring commitment to their strategic vision and operational goals.

**Example 7.** We are **committed** to conducting activities in a manner that promotes positive and open relationships with local communities.

**Example 8.** We are **committed** to becoming an international company of excellence.

Table 6 reveals a clear division within the high-frequency words of [Business: General]. These words fall into two primary groups: those related to enterprises themselves (e.g., "business," "company," "corporation") and those associated with commercial activities (e.g., "economy," "commercial," "stock exchange"). This distinction highlights the dual focus of this semantic domain, encompassing both the actors within the business landscape and the broader economic environment in which they operate. For further analysis, we will delve deeper into the high-frequency terms "business" and "economy" exploring their collocations and contextual usage to gain a more nuanced understanding of how Chinese enterprises construct their identities within this domain.

The frequent collocations of "business" reveal the primary scope of operations for Chinese companies in Africa, including traditional labor-intensive industries like infrastructure construction and mineral development, as illustrated in Example 9. Collocations such as "mining," "processing," mineral," and "engineering contracting," highlight the focus on resource development and infrastructure construction in Africa. This also indirectly reflects the relatively limited presence of emerging technology industries in the business operations.

**Example 9.** The Company engages in the non-ferrous metal industry, mainly the mining and processing **business**.

Furthermore, the frequent collocations of "economy" (e.g., "international," "world-class," "local," "global") illustrate the extensive geographic reach of these companies, highlighting their global operations and integration into both the global economy and local African communities as shown in Example 10. Moreover, concepts like "sustainable business" and "zero-emission economy" reflect that these companies are not only pursuing economic interests but also striving for sustainable development by considering social and environmental factors to ensure long-term business success and enhance their reputation and brand image while creating a more stable operating environment.

**Example 10.** It has also built many major infrastructure projects of rail transit, bridges and urban utility tunnels to serve the national **economy** and people's livelihood.

**6.1.2 Analysis of category 2.** Table 7 highlights high-frequency words and collocations from the S8 + [Helping] domain in Category 2, illustrating the role of Chinese enterprises in Africa as practical actors. This domain showcases their tangible contributions to Africa's

**Table 6. High-frequency words and collocations of semantic domain I2.1 in Category 1.**

| Semantic domain | High-frequency words | Example | Collocations |
|---|---|---|---|
| I2.1 | business(325),company(314),enterprise(123),corporation(71); | business (325) | mining and processing, mineral trading |
| | economy(41),commercial(24), contractors(24),stock exchange (17), listed companies (5) | economy (41) | world,worldwide;national,Chinese,regional, local; zero emission;digital |

**Table 7. High-frequency words and collocations of semantic domain S8 + in Category 2.**

| Semantic domain | High-frequency words | Example | Collocations |
|---|---|---|---|
| S8 + | services(162),cooperation(102),protection(95), benefits(37),welf-are(30),care(17),guidance(14), assistance(13),charity(13); support(90),promote(71),help (29),serve(24),protect(24) | support (90) | health,safety, education; the public,partners, customers,employees,women |

advancement and well-being through verb tokens such as "support," "promote," "help," "serve," "protect," and "encourage," and noun tokens like "services," "benefits," "welfare," "care," "guidance," "assistance," and "charity." These terms collectively underscore the commitment of companies to the progress and the well-being of African people, as exemplified in Example 11.

**Example 11.** CRBC is actively engaged in social public **welfare** and **charity** activities, and provides timely aids to the host countries government and people when disasters occur.

In addition to a general understanding of the frequent words in semantic domain S8 + [Helping], the high-frequency word "support" was examined carefully due to its function as both a noun and a verb. A closer examination of the 90 occurrences of "support" reveals that it appears as a noun 51 times and as a verb 39 times, indicating a preference for the noun form. Using "support" as a noun allows organizations to define their offerings more specifically (e.g., technical support, financial support, talent support), while using it as a verb implies an obligation. When used as a noun, "support" often appears in the structure "the support of + noun" as shown in Example 12 to highlight the source of support desired by the enterprise (e.g., "with the care and support of partners," "win support of the public," "win trust and support of customers"), clearly identifying the entities from which the company seeks backing. Another common collocation is "provide/render support for/to + noun," as shown in Example 13 emphasizing the recipient of the support. For instance, this structure is used in phrases like "provide technical support for the operation of projects," "provide stronger talent support for enterprise development," and "rendered generous support to the local government,"where the prepositions "for" and "to" introduce the beneficiaries of the support. This analysis reveals that companies tend to use "support" as a noun to identify both the sources and targets of support, precisely articulating their needs and offerings.

**Example 12.** With the care and **support** of partners around the world, it has now developed into an international engineering group.

**Example 13.** CRBC has proactively organized a variety of social welfare and charity activities, and rendered generous **support** to local government and people to deal with natural disasters or dangerous circumstances.

When "support" is employed as a verb, the subject often includes the first-person plural "we" or specific corporate entities, such as CCECC Nigeria. The direct objects following this verb frequently relate to domains like health, safety, education, and livelihood as shown in Example 14, target demographics such as women and working mothers as shown in Example 15, and expansive community endeavors like employee volunteer programs. There is a pronounced focus on bolstering local governance and community welfare as illustrated in Example 14, with comparatively less emphasis on supporting customers and employees. This suggests that while companies value recognition from local authorities, they may overlook the needs and entitlements of customers and staff. Such a pattern signals a strategic prioritization of external affirmation over comprehensive stakeholder engagement.

**Example 14.** We develop infrastructure, **support** health, safety, and education efforts.

**Example 15.** More than a decade ago, Lenovo recognized the need to **support** women in the workplace, and a small group of female executives created Women in LenovoLeadership (WILL) as its first ERG.

**Example 16.** CCECC Nigeria has proven to be a truly reliable partner willing and able to **support** the prosperity and development of Nigeria.

These terms under S8 + [Helping] signify not only the provision of products and services but also active participation in social welfare projects within African nations. By actively engaging in infrastructure development and initiatives promoting health, safety, and education, the companies showcase their dedication to contributing positively to the communities they serve. However, a more balanced approach is needed, encompassing substantive support

for customers and employees, ensuring their needs and rights are met alongside broader community endeavors.

**6.1.3 Analysis of category 3.** The third category, which encompasses semantic domain Z2 [Geographical names] and M7 [Places], underscores the significant role of Chinese enterprises in Africa as key players and competitors on the global scene. Table 8 presents the top twenty high-frequency words within the semantic domain of Z2 [Geographical names], further classifying these terms into three distinct types according to their implications and geographical specificity, as detailed in Table 9. This categorization facilitates a sufficient understanding of the geographical dynamics at play, offering insights into the strategic locations and international aspirations of Chinese enterprises operating across Africa. Through this analysis, the significant footprint and competitive stance of these companies on the international stage are brought to the forefront, reflecting their ambition and strategic positioning in the global economy.

In Table 8, "high-frequency words" refer to the words that appear most often within the semantic domain Z2 [Geographical Names] of the CCIA corpus, without comparison to a general or reference corpus.

Table 9 presents the classification of high-frequency words and collocations within the semantic domain Z2 [Geographical names]. The first type of high-frequency words within the Z2 [Geographical names] domain includes terms such as "China," "Chinese," "Hong Kong," "Republic of China," "Shanghai," "North China," and "Beijing." The predominant appearance of "China" (192) and "Chinese" (129) underscores an emphasis on the companies' origins and affiliations with China, marking a distinct identity as Chinese enterprises. This strong linkage to their homeland suggests a strategic attempt by Chinese businesses in Africa to capitalize on the favorable international image of China. Such an association is advantageous for garnering

**Table 8. Top 20 high-frequency words of semantic domain Z2 [Geographical names].**

| Rank | Word | Frequency | Rank | Word | Frequency |
|---|---|---|---|---|---|
| 1 | China | 192 | 11 | Kenya | 15 |
| 2 | Chinese | 129 | 12 | Tanzania | 15 |
| 3 | Nigeria | 86 | 13 | Shanghai | 14 |
| 4 | Africa | 74 | 14 | South Africa | 14 |
| 5 | African | 30 | 15 | North China | 13 |
| 6 | Asia | 22 | 16 | Beijing | 12 |
| 7 | Nigerian | 22 | 17 | Ghana | 11 |
| 8 | Hong Kong | 18 | 18 | Middle east | 10 |
| 9 | Europe | 17 | 19 | North America | 10 |
| 10 | Republic of China | 17 | 20 | America | 9 |

**Table 9. High-frequency words classification and collocations of semantic domain Z2 [Geographical names].**

|  | Z2 [Geographical names] | Collocations |
|---|---|---|
| 1 | China(192), Chinese(129), Hong Kong(18), Republic of China(17), Shanghai(14), North China(13), Beijing(12) | Chinese operation/efficiency/characteristics/National High Quality Engineering Prize |
| 2 | Nigeria(86), Africa(74), African(30), Nigerian(22), Kenya(15), Tanzania(15), South Africa(14), Ghana(11) | African employees/governments/women/users/students; Nigerian government/ universities/ Railway Modernization Project/ partners |
| 3 | Asia(22), Europe(17), the Middle East(10), North America(10), America(9) | Southeast Asia |

support and trust from local governments and communities, thereby facilitating the operations of companies and aiding in overcoming transient challenges.

Furthermore, references to "Beijing," "Shanghai," and "Hong Kong," highlight the significant commercial and financial influence wielded by these cities on the global stage. As economic powerhouses within China, they symbolize the country's robust economic hubs and international commerce centers, drawing considerable investments and business opportunities. Additionally, the frequent collocation of "Chinese" with terms such as "China' s Luban Prize" in Example 17 and "Chinese efficiency," "Chinese elements," in Example 18 reflects a confidence in the prowess of Chinese manufacturing. These references indicate that Chinese enterprises in Africa pride themselves on the efficiency and productivity emblematic of operations in China, highlighting the advanced status and leading role of Chinese manufacturing in the global market. This strategy not only showcases the unique features of products or services but also encapsulates a broader sense of cultural identity and affinity, underscoring the multifaceted identity of Chinese enterprises as global ambassadors of Chinese culture and manufacturing excellence.

**Example 17.** SEPCOIII has been awarded National High Quality Engineering-Gold Award for 7 times and **China**' s Luban Prize for 10 times.

**Example 18.** The integration of the new management revitalizes the business by bringing "**Chinese** efficiency and **Chinese** elements ".

The second type of high-frequency words within the Z2 [Geographical names] features names of African countries, such as "Ghana" and "Nigeria" in West Africa, "Kenya" and "Tanzania" in East Africa, and "South Africa," a leading economy on the continent. Through the use of these country names on their official websites, these enterprises showcase their extensive reach and profound impact across the African continent, thereby demonstrating the wide-ranging scope of their business operations.

The collocations with "African" such as "African employees," "African governments," "African women," "African users," and "African students" reflect the multifaceted engagement of Chinese enterprises with the continent. The example 19 shows, TRANSSNET, a Hong Kong-based company that develops and sells music application software and other products, leveraging a Chinese Internet technology company NetEase's tech and TRANSSION's African market presence for tailored Internet products. The example 20 illustrates a Chinese manufacturer of hair products, Henan Rebecca's appeal to African women with reliable, versatile goods, cementing its popularity. These examples highlight the adeptness of companies at meeting African market needs. Companies in Africa, by offering products and services that cater specifically to the needs of African users, thus strengthen the ties between China and Africa.

**Example 19.** Focusing on creating the most favorite Internet products for **African** users, TRANSSNET combines NetEases strong technical strength and advanced Internet product operation concepts with TRANSSION's mature channel resources and solid market foundation in Africa.

**Example 20.** By its stable supply of goods, changeful style, functional sex is strong of the original silk unique advantages of the millions of **African** women love become the first choice.

The examples 21 and 22 demonstrate the commitment of Chinese firms to fostering educational ties and investing in local talent by sponsoring African students and hiring graduates from Nigerian universities. These actions underline their dedication to contributing to local employment and educational development, showcasing Chinese enterprises' significant impact on Africa's socio-economic progress. Through such initiatives, these companies play a vital role in fostering mutually beneficial partnerships with African governments and citizens, contributing to their socio-economic upliftment.

**Example 21.** CJIC has sponsored a number of **African** students to pursue education in China.

**Example 22.** Every year, we sign employment contracts with fresh graduates from **Nigerian** universities, carry out induction training and follow-up appraisals for them, and train local staff with great efforts.

The examples 23 and 24 highlight the focus of China owned subsidiaries in Nigeria on building strategic political relationships with African governments, particularly in the promotion of digitalization and infrastructure projects. These partnerships are crucial for facilitating business operations and accessing local markets, showcasing the importance of political ties in the success of Chinese enterprises in Africa. In Example 24, the mention of "Grade-A qualifications" positions CCECC Nigeria as a trusted contractor endorsed by the Nigerian government. This designation not only highlights CCECC Nigeria's reliability but also showcases its pivotal role in infrastructure development and economic growth initiatives. Furthermore, it signifies CCECC Nigeria's commitment to technical excellence and high-quality standards, reinforcing its reputation as a reputable and dependable partner in Africa's development landscape.

**Example 23.** StarTimes began to expand its business to Africa in 2002, and has been working closely with **African** governments to jointly promote digitalization and informatization.

**Example 24.** As the largest construction contractor in Nigeria and even in West **Africa**, CCECC Nigeria has obtained Grade-A qualifications for constructing projects of the **Nigerian** government.

The third type of high-frequency words, such as "Asia,"" "Europe," " "the Middle East," "North America," and "the United States," demonstrate the expansive reach of Chinese enterprises beyond Africa and their home base. This global presence underscores their significance as major global players, wielding influence and competitive strength across diverse markets.

Table 10 reveals how Chinese enterprises navigate and articulate their identities in a globalized context through an analysis of M7 [Places].The high-frequency words within this semantic domain fall into two distinct categories: global/international context and national/local context. The global/international context, encompassing terms like "international," "countries," and "foreign," emphasizes engagement with various markets and roles in the international arena. This reflects themes of globalization and cross-border relationships, highlighting how companies position themselves globally. Collocations of "international" (e.g., "presence," "benchmarks," "practices") further illustrate this global engagement. For instance, CMOC's "growing international presence"in Example 25 emphasizes its expanding global influence, particularly in Africa, while its focus on "sustainable development" reinforces its commitment to global standards. Similarly, "aligning with leading international benchmarks" in Example 26 highlights a commitment to meeting or exceeding global standards, enhancing credibility and demonstrating dedication to sustainability and responsiveness to global expectations.

**Example 25.** As a company with a growing **international** presence, CMOC fully recognizes the importance of sustainable development.

**Table 10. High-frequency words and collocations of semantic domain M7 [Places] in Category 3.**

| Semantic domain | High-frequency words | Example | Collocations |
|---|---|---|---|
| M7 | international(264), countries(131), foreign(51); | international (264) | presence, benchmarks, practices |
| | national(69), regions(64), province(38), cities(12); | national (69) | regulations, strategy, economy, security |

**Example 26.** We will place ESG at the core of our management by developing a systematic ESG management framework, aligning with leading international **benchmarks**.

In contrast, the national/local context uses terms like "national," "regions," "province," and "cities" to underscore connections to the company's home country and local environments. This reflects national identity and the importance of local economic and social dynamics, emphasizing commitment to regional development and national priorities. Collocations of "national" (e.g., "regulations," "development," "strategy," "priorities," "economy," "security") convey themes related to governance, policy, and national identity. "National regulations" specifically highlight a framework for legal and ethical compliance. The example 27 illustrates HE's commitment to national economic development and stability, while the Example 28 uses "the Chinese Dream of national rejuvenation" to encapsulate a larger cultural and political aspiration, linking company success with national goals and framing achievements as part of a collective national effort.

**Example 27.** Now HE has become one of the 52 state-owned backboneenterprises that concern **national** security and the lifelines of the **national** economy.

**Example 28.** It pursues to become the top brand in global investment and construction arena and a banner of China's reform, development and urbanization aimed to realize the Chinese Dream of **national** rejuvenation.

By aligning their corporate identity with national interests, these enterprises assert their significance in global markets and position themselves as contributors to both local and home country development. This dual focus allows them to navigate the complexities of a globalized world while maintaining strong ties to their national identity. The analysis of M7 [Places] provides valuable insights into how Chinese enterprises strategically use language to construct their identities and communicate their commitment to both global engagement and national development.

**6.1.4  Analysis of category 4.**  The fourth category as demonstrated in Table 11, featuring N1 [Numbers], distinctly highlights the leadership roles and pioneering status of Chinese enterprises within the industry, reinforcing their identities as trailblazers and industry frontrunners. The exploration of the semantic domain N1 [Numbers] and its leading high-frequency words and collocations illustrated in Table 11, such as "years" "one," "million," "billion," and "zero," unveils a notable pattern in their usage.

The examples 29 and 30 strategically utilize the semantic domain N1 [Numbers] to highlight extensive corporate histories. This choice underscores the companies' longevity and commitment to their sectors and regions of operation. In traditional Chinese culture, the historical depth of a company symbolizes resilience, consistency, and trustworthiness. By emphasizing foundational dates or durations of engagement in the African market, these companies position themselves as seasoned and stable entities with a long-term commitment to their operations.

**Table 11.  High-frequency words and collocations of semantic domain N1 [Numbers].**

|  | Words | Collocations |
|---|---|---|
| 1 | 1950,1958,1969... | In+year |
| 2 | one(71) | one of the highest/most prominent/most powerful |
| 3 | million(28),billion(21) | RMB/USD+number+million/billion |
| 4 | zero(15) | hunger,injuries,accidents;<br>defects;<br>pollution,emission; |

**Example 29.** Behind every great company, there is a great story. Hisense S.A. (PTY) Ltd. entered the South African market in **1996**.

**Example 30.** CMOC Group Limited (CMOC or the Company) was founded in **1969** and has since completed two respective mixed ownership reforms in 2004 and 2014.

The examples 31 and 32 employ the structure "one of + superlative adjective," where terms like "one of the most successful" and "one of the pioneering" are used. This pattern asserts superiority and distinction, highlighting the companies' professionalism, leadership, achievements, and capacity for innovation within their respective industries. Through the strategic use of superlatives and positive adjectives, these companies shape an identity that emphasizes their prominence and excellence in the field.

**Example 31.** This Chinese company providing construction and engineering services is **one** of Nigeria's most successful Chinese-owned companies.

**Example 32.** As **one** of the pioneering Chinese companies entering the international market, CCECC is now developed into a large-scale state-owned enterprise.

In Example 33 and 34, the use of "million" and "billion" enables Chinese enterprises in Africa to highlight their substantial financial contributions and initiatives. By outlining investments in philanthropy, business projects, community infrastructure, and environmental conservation, these companies effectively demonstrate their significant financial capacity and commitment to social and environmental welfare. Such disclosures serve multiple purposes: firstly, they underscore the enterprise's financial strength, reflecting its stability and reliability as a business partner. Secondly, by emphasizing their involvement in charitable endeavors and development projects, these companies position themselves as conscientious entities dedicated to positively impacting communities and environments beyond profit-making. This portrayal positions them as leaders in their industries and responsible corporate citizens committed to societal betterment.

**Example 33.** In 2022, CMOC spent around RMB 379 **million** on environmental protection.

**Example 34.** Since 2017, we have signed 288 subcontracting contracts with 170 Nigerian companies with a total contract value of approximately 25 **billion** naira (around USD 82 million) Enhancing Local Procurement.

Within the semantic domain N1 [Numbers], the high-frequency word "zero" is particularly striking for its association with ambitious goals and commitments, as seen in concordances like "zero hunger," "zero injuries," "zero accidents," "zero defects," "zero pollution," and "zero emission economy." The example 35 highlights the dedication of companies to employee safety through its pursuit of zero injuries and accidents, reflecting Chinese values of prioritizing people's well-being.

**Example 35.** We have enhanced employees safety capabilities, pursued the safety management goal of **zero** injuries and **zero** accidents.

The examples 36 and 37 demonstrate the commitments of Chinese companies to excellence and environmental stewardship. By adhering to principles like "people-centered, quality foremost, safety first, environment prioritized" and striving for "zero defects, zero injuries, and zero pollution," they establish themselves as entities with rigorous standards. The pursuit of a "zero emission economy" reflects their proactive stance on environmental preservation and their efforts to minimize ecological footprints. These initiatives showcase their dedication to contributing to a sustainable future, reinforcing their identity as advocates for environmental responsibility and quality excellence.

**Example 36.** We are always committed to the QHSE principle of "people-centered, quality foremost, safety first, environment prioritized" to achieve "**zero** defects, **zero** injuries and **zero** pollution".

**Example 37.** We are proud of the encouraging progress we have made that helps accelerate the global transition to a net **zero** emission economy.

The strategic deployment of "zero" within the N1 [Numbers] symbolizes a comprehensive dedication to employee welfare, product quality, and environmental sustainability. This nuanced portrayal reinforces the company identities as both industrial leaders and responsible members of the global community, committed to fostering well-being, quality, and sustainability. Together with figures like "years," "one," "millions," and "billions," they depict Chinese enterprises as key players on the world stage, capable of influencing markets, shaping industry trends, and significantly contributing to global economic dynamics.

**6.1.5  Analysis of category 5.** The fifth category, encompassing Semantic domain S5 + [Groups and affiliation], delves into the dynamics of organizational and community relationships, epitomizing the identity of Chinese enterprises in Africa as trustworthy partners and responsible members in the society. The lexicon under S5 + includes terms such as "group(173)," "community(81)," "society(80)," "team(60)," "public(48)," "chain(42)," "network(37)," and "together(23)," which can be segmented into three specific types based on their contextual implications.

The first type, featuring "group," "chain," and "network," showcases the mature business chain and comprehensive management model. The examples 38, 39, and 40 illustrate the maturity and sophistication of the business models of companies. The terms like "group," "network," and "chain" highlight their comprehensive management structures and interconnected operations. These terms suggest the companies' capability to efficiently manage complex networks and value chains, contributing to their competitive advantage in the marketplace. This emphasizes not only the depth of their organizational structure but also their ability to establish intricate connections and relationships, both locally and globally.

**Example 38.** CSCEC is now a global investment and construction **group** featuring professional development and market-oriented operation.

**Example 39.** His group focuses on creating a supportive **network** of moms who help foster a smooth transition into the world of balancing work and motherhood.

**Example 40.** SEPCO has the integrated service capabilities of the entire industrial **chain** of power, oil...

The examples 41, 42, and 43 imply the proactive social responsibility of Chinese enterprises in Africa. The terms like "community," "society," and "public" underscore their constructive relationships with local populations. Phrases associated with "community" indicate deep integration into local communities, portraying these enterprises as proactive agents of social welfare and advancement. This highlights their dedication to contributing resources and efforts towards the well-being of local communities, both at home and abroad.

**Example 41.** Adhering to the idea of full cooperation, active service, reciprocating **society**, CRBC is actively engaged in social public welfare and charity activities.

**Example 42.** SEPCO also pays great attention to contributing **public** welfare activities at home and abroad.

**Example 43.** We continue our engagement with and investment in **communities** affected by our mining operations.

The examples 44 and 45 emphasize the significance of collaboration and collective effort among employees, reflecting the companies' commitment to fostering transparent and inclusive work environments. The terms like "team" and "together" underscore their role as supportive and equitable employers, promoting unity, mutual respect, and shared responsibility. By prioritizing teamwork, they cultivate cultures of openness and cooperation, valuing ideas and contributing to both individual growth and collective success.

**Example 44.** With the improvement of production capacity, the staff **team** has been tempered in the construction of world-class projects.

**Example 45.** HE members work and act **together** and in a concerted manner, to jointly develop HE and boost the great cause of HE.

Chinese enterprises in Africa solidify their presence as deeply committed entities in cultivating positive and sustainable relationships through their approach to groups and affiliations. Whether through fostering international cooperation, engaging in community development, or promoting a collaborative workplace culture, these enterprises showcase their reliability and engagement as partners. This reinforces their credibility and trustworthiness while reaffirming their dedication to making positive contributions to socio-economic landscapes.

**6.1.6  Analysis of category 6.**  In the final category, Z8 [Pronouns], enterprises utilize pronouns as essential tools for self-referential communication. Pronouns serve as crucial markers of identity, representing the most direct form of address within corporate discourse [34]. They effectively express and modulate social relations, hierarchical standings, and power dynamics. Pronouns like "we(895)," "our(671)," "its(312)," and "it(187)" play distinct roles in the communication strategies of enterprises.

The examples 46 and 47 showcase a preference for inclusive pronouns like "we" and "our," emphasizing the focus within the internal team and excluding external stakeholders like customers or shareholders. This usage highlights corporate autonomy and a well-defined structure, reflecting China's high power distance culture [35,36]. This cultural framework prioritizes clear distinctions in social status, management hierarchies, and authority, mirrored in the selective application of pronouns to denote a collective internal identity.

**Example 46. We** are ready to seek greater growth around our strategic goals in key regions and key products.

**Example 47.** World-class projects that are launched to expand **our** production capacity.

In contrast, pronouns like "its" and "it" are employed to represent the company in a formal and detached manner. The examples 48 and 49 utilize these pronouns to introduce objectivity and neutrality in the firm's external communications. This approach is crucial for interacting with various partners, including customers, shareholders, and the general public. The neutrality and formality conveyed through these pronouns are essential for establishing trust and credibility, presenting the company as an impartial and professional entity.

**Example 48.** The sustainable business practices will also help CMOC to secure **its** long-term success.

**Example 49. It** was listed on Hong Kong Exchanges (HKEX: 03993) in 2007.

In summary, the strategic use of pronouns in referring to the company serves various objectives, such as maintaining a formal tone, asserting corporate authority, and delivering objective communication. These linguistic choices present enterprises as formal entities with a clear sense of direction and authority, skillfully navigating the complexities of social relationships and power structures in the corporate realm.

**6.1.7  Distilled corporate identities.**  This study reveals the diverse identities that Chinese enterprises construct in Africa by analyzing the semantic domains related to their activities. Initially, these identities include roles as professional service and product providers, bearers of technical excellence and high-quality standards, key global players, reliable partners, catalysts for economic growth, transparent entities, industry front-runners, stable organizations, proactive actors, innovative entities, advocates for environmental preservation, committed philanthropists, talent nurturers, equitable employers, and global ambassadors of Chinese culture. To streamline understanding and eliminate redundancy, these identities can be consolidated into five principal categories: leading providers of professional services and high-quality products; global influencers driving innovation and economic growth; champions of

environmental preservation and philanthropy; advocates for inclusive workplaces and talent development; and promoters of Chinese culture on the global stage.

## 6.2 Identity construction features

**6.2.1 Relationship between corporate identity and communication strategies.** To assess the relationship between corporate identity and communication strategies, we developed a scoring system for Corporate Social Responsibility (CSR) and Corporate Ability (CAb) strategies based on specific indicators across company websites. Each indicator was assigned a binary score (1 for presence, 0 for absence), reflecting the focus of each enterprise's communication strategy. Table 1 show the specific CSR and CAb indicators used in the analysis. Two trained coders independently reviewed each website and assigned scores to each indicator. Any disagreements between coders were resolved by a third evaluator. Inter-rater reliability, assessed using Cohen's kappa coefficient, was high (0.85), indicating strong agreement between coders. For example, the China Road and Bridge Corporation (CRBC) received a CSR score of six and a CAb score of three, suggesting a CSR-focused strategy. Tables 12 and 13 provide further details of the annotation process.

This rigorous methodology allowed for a quantitative assessment of the relationship between corporate identity and communication strategies, revealing how companies prioritize different aspects of their identity through their online communication. The use of multiple coders and a third evaluator ensured the reliability and consistency of the scoring process, strengthening the validity of the analysis.

Our preliminary analysis of language data has enabled us to identify key components of communication strategies prioritized by companies. This analysis also illuminates how Corporate Social Responsibility (CSR) and Corporate Ability (CAb) strategies intertwine to shape corporate identity. Each component is systematically categorized under its corresponding identity, as outlined in Tables 14 and 15. This detailed classification process allows us to ascertain the dominant identity types that businesses aim to cultivate. Additionally, the relationship between these components and the broader corporate identities not only clarifies the roles of these communication strategies but also highlights their integral role within the comprehensive corporate identities.

**6.2.2 Identity construction features in CAb enterprises.** Fig 1 reveals that among the 200 enterprises surveyed, a significant majority of 156 (78% of the sample) have adopted a Corporate Ability (CAb) strategy. In contrast, 21 enterprises, accounting for 10%, have opted for a Corporate Social Responsibility (CSR) strategy, while 23 enterprises, representing 12%, have implemented a hybrid strategy. This highlights the predominant adoption of

**Table 12. CAb scores of CRBC.**

| Indicators of CAb strategy | Annotation evidence | Score |
|---|---|---|
| Expertise in product or service quality | CRBC is one of the four large State-owned companies in China that earliest entered into the international engineering contracting market. | 1 |
| Global success | CRBC has established relevant branches in nearly 60 countries and regions spreading from Asia, Africa to Europe and America. | 1 |
| Implementation of quality control program | Not found | 0 |
| Industry leadership | China Civil Engineering Zhan Tianyou Prize<br>China National Engineering Luban Prize (Overseas Project) | 1 |
| Market orientation | Not found | 0 |
| Innovation and R&D efforts | Not found | 0 |
| Total | – | 3 |

**Table 13. CSR scores of CRBC.**

| Indicators of CSR strategy | Annotation evidence | Score |
|---|---|---|
| Environmental stewardship | CRBC strictly abides by the local laws and regulations on environment protection, and integrates the idea of environment protection into all links of project planning, management and construction. | 1 |
| Philanthropic contribution | CRBC is actively engaged in social public welfare and charity activities, and provides timely aids to the host countries' government and people when disasters occur. | 1 |
| Educational commitments | CRBC has always been concerned with the educational development in the host countries and gives priority to reciprocating the local education by assisting host countries' students to come to study in China | 1 |
| Employee involvement | We full safeguard employee' rights to know, supervise and participate in decision-making, build channels for good communication between Chinese and local employees, and give full play to the role of employees in democratic management. | 1 |
| Public health commitments | CRBC is always been caring for staff health and devoting to providing healthy, safe and humane working and living environments. | 1 |
| Sponsorship of cultural activities | Various cultural exchanges enable both Local and Chinese staff to relax in their spare time, develop healthy living and working habits, and deepen friendship. | 1 |
| Total | – | 6 |

**Table 14. CSR strategy based identities.**

| | Indicators of CSR strategy | Corporate focused identities |
|---|---|---|
| 1 | Environmental stewardship | Advocates for environmental preservation |
| 2 | Philanthropic contribution | Committed philanthropists |
| 3 | Educational commitments | Sponsors and supporters of talent nurturing |
| 4 | Employee involvement | Equitable, supportive, and inclusive employers |
| 5 | Public health commitments | Guardian of staff wellness |
| 6 | Sponsorship of cultural activities | Global ambassadors of Chinese culture |

**Table 15. CAb strategy based identities.**

| | Indicators of CAb-focused strategy | Corporate focused identities |
|---|---|---|
| 1 | Expertise in product or service quality | 1) Professional service and product providers, 2) Bearers of technical excellence and high-quality standards |
| 2 | Global success | 1) Key global players, competitors, and contributors 2) Strategic, reliable and esteemed partners 3) Catalysts for economic growth |
| 3 | Implementation of quality control program | 1) Bearers of technical excellence and high-quality standards 2) Transparent, integrated, and formal entities |
| 4 | Industry leadership | 1) Trailblazers and industry front-runners, 2) Mature and stable entities, |
| 5 | Market orientation | 1) Proactive agents and actors |
| 6 | Innovation and R&D efforts | 1)Dedicated, visionary and innovative entities |

CAb strategies, leading to noticeable differences in performance scores among the surveyed enterprises.

This strategic orientation can effectively communicate essential information about the enterprise's capabilities to customers and shareholders. Further analysis within the subgroup of 156 enterprises shows that 81 of them (approximately 52%) have CAb scores between 3 and 4, suggesting a moderate level of corporate capability utilization. These enterprises are proficient in communicating important information to their stakeholders. Additionally, 46 enterprises (approximately 29% of the total) attained a CAb score ranging from 5 to 6, indicating

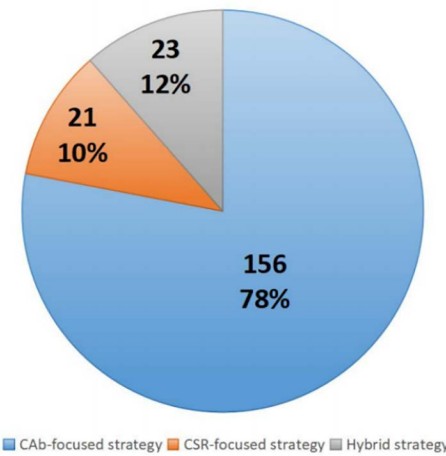

■ CAb-focused strategy ■ CSR-focused strategy ■ Hybrid strategy

**Fig 1. The adoption of corporate communication strategy.**

a high level of proficiency in utilizing their corporate capabilities. Conversely, 29 enterprises (approximately 19%) scored between 1 and 2 on the CAb scale, implying a relatively lower level of capability utilization, as outlined in Fig 2. This subgroup may encounter challenges in effectively disseminating crucial information to their target audience, reflecting potential areas for improvement in communication and operational strategies.

The data reveals that a substantial number of enterprises, specifically 81, commonly score between 3 and 4. These firms are particularly strong in areas such as product or service quality, global success, industry leadership, and research and development efforts. The focus on these dimensions suggests a deliberate attempt to craft corresponding corporate identities. The companies emphasize their expertise in providing high-quality products or services, positioning themselves as "professional service and product providers" and "experts with technical excellence and high-quality standards." In corporate profiles, the initial information presented typically includes foundational details like the establishment year, business nature, and product range. This sequence not only answers the fundamental question of "Who am I?" "What I do?" but also strategically aligns with the intent to shape a distinct corporate identity. Furthermore, these companies highlight their global engagement, presenting themselves as "key global players and reliable partners." They also stress their leadership in the industry, portraying themselves as "influential industry leaders and pioneers" with substantial financial robustness. Lastly, they underscore their commitment to innovation and R&D, positioning themselves as "innovators and trendsetters" and "dedicated, forward-thinking organizations."

These strategic emphases in corporate profiles are designed to establish a multifaceted corporate identity that reflects expertise, international success, industry leadership, and a commitment to innovation. These identities not only communicate the current status of enterprises but also help in forging robust connections with stakeholders, shaping how these companies are perceived in the global marketplace.

The analysis of the data presented reveals that among the Chinese enterprises in Africa, only 29 percent scored between 5 and 6 on the assessment scale, indicating their ability not only to establish robust corporate identities but also to effectively communicate insights about the implementation of quality control programs and market-oriented practices. These high scores denote these enterprises as "formal and principled organizations" as well as "honest and authoritative organizations," thus successfully differentiating them and enhancing their stature and goodwill in the public eye. The strategic implementation of these practices sets

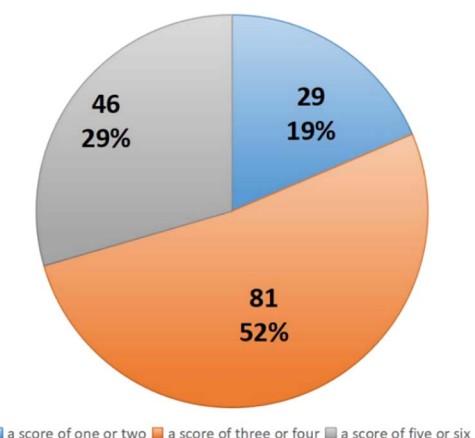

**Fig 2. The overall scores of CAb focused enterprises.**

these organizations apart from other industry players in the region, bolstering their public image, credibility, and overall reputation.

Conversely, only 19% of the enterprises achieved a score of 1 or 2, underscoring a limited emphasis on proficiency in product or service quality and global success—attributes that are frequently highlighted on nearly every company website. This disparity suggests a prevalent aspiration among companies to be recognized as "providers of professional services and products" and "global enterprises."

**6.2.3 State-owned vs. private enterprises.** The analysis of score statistics suggests that enterprises with a CSR strategy demonstrate less variability in their scores. The variations in scores and underlying intentions for identity construction may stem from the unique nature of the enterprises. There is no significant disparity in CAb scores between state-owned and private enterprises. However, differences are apparent in CSR scores, warranting a detailed exploration of identity construction characteristics based on CSR scores for state-owned versus private enterprises.

The contrasting roles of state-owned and private enterprises significantly shape their constructed identities. In this study encompassing 200 enterprises, 80 are state-owned and 120 are private. State-owned enterprises collectively attained a total CSR score of 858, while private enterprises scored 861 in total. Notably, 21.5% of state-owned enterprises did not address any CSR strategies, compared to 40.4% of private enterprises. Additionally, 13.8% of state-owned enterprises achieved scores of 5 and 6, compared to 9.8% among private enterprises, indicating a stronger emphasis on CSR strategies among state-owned entities to cultivate a responsible societal identity.

The variation in scoring among enterprises stems from several factors. Firstly, state-owned enterprises (SOEs) integrate CSR initiatives with national agendas, focusing on poverty alleviation, environmental conservation, and community development. Their tangible impacts reinforce their identity as responsible corporate citizens, contributing to sustainable socio-economic development. Wei and Wu note that SOEs often align their identity with national development goals, positioning themselves as champions of the state's socio-economic vision [37]. Secondly, mature communication systems enable effective dissemination of positive CSR messages. Fu and Zhu highlight that Chinese SOEs often employ hierarchical messaging to maintain a consistent and credible corporate identity [38]. At last, SOEs benefit from governmental support, including financial backing and policy incentives, enhancing their competitive edge and enabling substantial projects.

In brief, state-owned enterprises demonstrate a stronger commitment to CSR, aligning their initiatives with national development priorities and leveraging governmental support to drive impactful projects. Their emphasis on social responsibility not only reinforces their corporate citizenship but also enhances their competitive advantage and market presence. In contrast, private enterprises exhibit greater variability in CSR engagement, potentially reflecting diverse business strategies and market pressures..

## 6.3 Socio-cultural factors analysis

The identity development of Chinese enterprises in Africa is shaped by socio-cultural factors, reflecting a complex interaction between Chinese cultural norms and the distinct African context.

In many Chinese enterprises, there exists a strong emphasis on showcasing industry leadership, which is deeply rooted in the cultural concept of "face." This concept entails projecting prestige and maintaining a notable external presence, often achieved through received awards, prizes, accolades, and rankings. For instance, SEPCO, a Chinese engineering and construction company, exemplifies this trend, prominently displaying its qualifications and honors on its official website. These accolades range from National Level Awards to International Awards, such as the "Best Construction Projects" prize and the "Outstanding Concrete Structure" prize in India. Furthermore, these companies frequently leverage recognition and positive feedback from various stakeholders, including society, users, and governments, to validate their reputation. This practice of seeking external validation aligns closely with China's cultural inclination towards saving face. For instance, CMOC, the largest molybdenum producer in Mainland China and among the world's largest companies in the field, highlights how its contributions have been widely recognized by society and emphasizes receiving the "2019 Henan Provincial Award for Poverty Alleviation and Dedication" from the Henan Provincial Government as a testament to its efforts. The intertwining of industry accolades, societal recognition, and governmental awards reflects a broader cultural ethos in Chinese enterprises, where maintaining a prestigious image and validating reputation are integral components deeply ingrained in the concept of "face."

The identity construction of Chinese enterprises in Africa reflects the commitment of China to preserving face and collective cultural values. Grounded in China's collectivist culture, these enterprises strategically leverage their Chinese origins to reinforce their identity. For example, CCECC, China Civil Engineering Construction Corporation Ltd, underscores its establishment under the approval of the State Council of the People's Republic of China in 1979. Moreover, they actively respond to national policies even while expanding overseas. SINOSURE, as a national policy-oriented insurance agency, contributes to initiatives like the "Made in China 2025" Initiative. Similarly, companies like China Unicom align their goals with China's "14th Five-Year Plan," particularly in digital infrastructure development. They position themselves as contributors to China's national rejuvenation, or the "Chinese Dream." Companies such as China State Construction Engineering Corporation, Bank of China, and CGCOC Group express dedication to realizing this vision, aligning their African efforts with China's broader goals. Overall, the identity construction of Chinese enterprises in Africa is a multifaceted process that emphasizes their origins, alignment with national policies, and commitment to China's overarching national rejuvenation narrative.

The identity construction process of Chinese enterprises in Africa also reflects China's human-centric values. These companies tackle global challenges in areas such as human rights, environmental governance, peace advocacy, and economic assistance. They notably prioritize support for vulnerable groups, with initiatives like Jereh Charity Fund of Yantai Charity Federation and Boji Relief Fund in Laishan providing aid to the elderly, children,

students, disabled individuals, and patients. CMOC, as seen on its website, commits to addressing poverty, inequality, climate change, environmental degradation, and other issues in collaboration with stakeholders, embodying China's people-centered values. Their dedication to initiatives beyond profit underscores China's emphasis on human welfare above all else.

The historical ties between China and Africa, marked by enduring alliances and collaborative endeavors like the Belt and Road Initiative and the Forum on China-Africa Cooperation, underscore the pivotal role of governmental support and cooperative frameworks in advancing Chinese enterprises in Africa. For instance, Sentuo Oil Refinery Limited and China-Africa Agriculture Investment Co., Ltd, are products of strategic initiatives like the Belt and Road Development Strategy. China-Africa Agriculture Investment Co., Ltd, jointly contributed by China National Agricultural Development Group Corporation and China-Africa Development Fund. CRBC, guided by China's "go global" strategy and the Belt and Road Initiative, has capitalized on these frameworks to enhance business expansion and core competitiveness. This holistic understanding of socio-cultural dynamics and historical contexts enriches the nuanced identity formation of Chinese enterprises in Africa, fostering deeper mutual understanding and cooperation between the two regions.

## 7. Conclusion

This investigation has underscored the significant role that discourse plays in constructing and disseminating the corporate identities of Chinese enterprises operating in Africa. Through an analysis of corporate communications, it was revealed that these entities strategically deploy a variety of discursive practices to foster a favorable identity and to strengthen their positions within African markets. These practices are not only pivotal in portraying their contributions to economic development and community engagement but also crucial in how they address challenges and integrate into the local social fabric.

The findings indicate that portraying Chinese corporations as both economic powerhouses and culturally adaptive entities allows them to effectively navigate complex public narratives. The adoption of corporate communication strategies that emphasize both Corporate Ability (CAb) and Corporate Social Responsibility (CSR) illustrates a sophisticated approach to balancing business objectives with social commitments.

In conclusion, the corporate identities of Chinese enterprises in Africa are dynamically constructed through discursive means that engage deeply with the cultural, economic, and social landscapes of the continent. This research contributes to the academic discourse on corporate identity construction in cross-cultural settings and provides practical insights for corporations aiming to enhance their global business strategies through effective communication.

## Supporting information

**S1_File. The corpus of Chinese Enterprises in Africa (CCIA).** This file contains a comprehensive collection of data on 200 Chinese enterprises operating in Africa, including sections such as "About Us," "Corporate Responsibility," and "Culture."
(PDF)

**S2_File. The details of CSA and CSB scores and company types for 200 Chinese enterprises operating in Africa.** This file provides a comprehensive overview of the scoring metrics and categorization of these enterprises, facilitating a better understanding of their performance and operational characteristics in the African market.
(XLSX)

**S3_File. Raw data from the Wmatrix on the Chinese Enterprises in Africa (CCIA) corpus.** This file contains the frequency and relative frequency of each semantic domain, as well as the words categorized within these domains.
(XLSX)

## Acknowledgments

We would like to express our sincere gratitude to the editors and the anonymous reviewers for their constructive comments on earlier versions of this paper. Any remaining errors are on our own.

## Author contributions

**Conceptualization:** Wenyu Liu.

**Data curation:** Yu Yang.

**Investigation:** Wenyu Liu.

**Methodology:** Yu Yang.

**Project administration:** Wenyu Liu.

**Resources:** Wenyu Liu.

**Supervision:** Wenyu Liu.

**Writing – original draft:** Yu Yang.

**Writing – review & editing:** Wenyu Liu.

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
