## [Decision Letter · Decision Letter 0]

7 Oct 2024

PONE-D-24-21342Exploring the narrative landscape: the discursive construction of identity for Chinese enterprises in AfricaPLOS ONE

Dear Dr. Liu,

Thank you for submitting your manuscript to PLOS ONE. After careful consideration, we feel that it has merit but does not fully meet PLOS ONE’s publication criteria as it currently stands. Therefore, we invite you to submit a revised version of the manuscript that addresses the points raised during the review process.

We look forward to receiving your revised manuscript.

Kind regards,

Muhammad Afzaal, PhD

Academic Editor

PLOS ONE

Journal Requirements:

2. We note that your Data Availability Statement is currently as follows: “All relevant data are within the manuscript and in Supporting Information files.”

Please confirm at this time whether or not your submission contains all raw data required to replicate the results of your study. Authors must share the “minimal data set” for their submission. PLOS defines the minimal data set to consist of the data required to replicate all study findings reported in the article, as well as related metadata and methods (https://journals.plos.org/plosone/s/data-availability#loc-minimal-data-set-definition). For example, authors should submit the following data: - The values behind the means, standard deviations and other measures reported; - The values used to build graphs; - The points extracted from images for analysis. Authors do not need to submit their entire data set if only a portion of the data was used in the reported study. If your submission does not contain these data, please either upload them as Supporting Information files or deposit them to a stable, public repository and provide us with the relevant URLs, DOIs, or accession numbers. For a list of recommended repositories, please see https://journals.plos.org/plosone/s/recommended-repositories. If there are ethical or legal restrictions on sharing a de-identified data set, please explain them in detail (e.g., data contain potentially sensitive information, data are owned by a third-party organization, etc.) and who has imposed them (e.g., an ethics committee). Please also provide contact information for a data access committee, ethics committee, or other institutional body to which data requests may be sent. If data are owned by a third party, please indicate how others may request data access.

4. Please upload a copy of Supporting Information Figure 1 and 2 which you refer to in your text on page 44.

Reviewers' comments:

Reviewer's Responses to Questions

**Comments to the Author**

1. Is the manuscript technically sound, and do the data support the conclusions?

Reviewer #1: Yes

Reviewer #2: Yes

2. Has the statistical analysis been performed appropriately and rigorously? 

Reviewer #1: N/A

Reviewer #2: Yes

3. Have the authors made all data underlying the findings in their manuscript fully available?

Reviewer #1: Yes

Reviewer #2: Yes

4. Is the manuscript presented in an intelligible fashion and written in standard English?

Reviewer #1: Yes

Reviewer #2: Yes

5. Review Comments to the Author

Reviewer #1: The authors have provided a relatively clear definition and theoretical background for "Corporate Identity" at the beginning of the literature review. However, the discussion on how scholars approach corporate identity from various perspectives could benefit from further elaboration. The statement, "Despite the extensive research on corporate identity across diverse fields such as economics, international relations, and journalism, there is a noticeable shortfall in applying discourse analysis methods to corporate identity studies," is too brief to capture the breadth of existing research. Offering more detail on how different researchers examine this concept across disciplines would help readers better understand the research context.

Similarly, the discussion of Kim and Rader’s corporate communication strategy typology requires further elaboration. I recommend incorporating more detailed information about the development background, theoretical underpinnings, and diverse applications of this typology in existing literature. This will provide readers with a clearer understanding of the theory's strengths and its relevance to your study.

In Section 5.1 Data Collection, it would be beneficial to include the specific methods used for data collection, such as whether the data was gathered through web crawling or manual copying and pasting. Additionally, presenting the data in a tabular format would enhance clarity. For example, you could include the number of texts and tokens collected from different types of corporates.

The sentence " delineates a primary identity unique to professional service and product providers, setting them apart from other sectors" is unclear. I recommend rephrasing it for clarity.

There are two concerns regarding the data presented in Table 4. First, although Category 1 includes the semantic domains A1.1.1 and I2.1, Table 4 only displays high-frequency words from the A1.1.1 domain. Please clarify whether the omission of data related to I2.1 was intentional or if it was an oversight in the "semtag" column. Second, please specify whether the terms "projects," "operation," and "production" are the highest frequency words within this semantic domain and provide their respective frequencies. If these terms are not the highest frequency words, please explain why they were selected for analysis.(Similarly, frequencies for high-frequency words in the other categories should also be included.

It would be helpful to provide a brief explanation of the calculation method used for the relative frequency presented in Table 2.

The analysis of high-frequency words in Section 6.1.2 could be more in-depth. A more thorough examination of specific verbs would be advantageous. For instance, analyzing the object of the verb "support" and the particular areas in which Chinese firms have provided assistance would enhance the depth of your analysis.

Table 5 is titled "Top 20 Keywords of Tagset Z2 [Geographical Names]." Generally, the term "keyword" denotes words that occur more frequently in the focus corpus compared to a general language or reference corpus. However, it appears that the use of "keywords" in your title may not align with this conventional definition. Could you please clarify how "keywords" is defined and applied in this context?

It appears that the methodology for assigning CSR and CAB scores to each company in Section 6.2.1 is not fully clear. I recommend offering a more detailed explanation and a specific example to clarify this process. For instance, how is a single CSR or CAB score determined for multiple texts collected from each company's website? Additionally, reporting on the reliability of the scores would be valuable. If the scoring involved multiple evaluators, including information on inter-rater reliability，such as Cohen's kappa coefficient， would strengthen the robustness of your analysis.

Reviewer #2: The study entitled “Exploring the narrative landscape: the discursive construction of identity for Chinese enterprises in Africa” examines the strategic employment of language in order to construct corporate identities and communication practices of Chinese companies in Africa. This study is a significant contribution as it provides an extensive comprehension of discursive identity construction of companies in cross-cultural context. Overall, this research article flows logically and the research undertaken is contextualized clearly. The introductory section offers relevant background of the study, highlighting the rationale and significance of this research. The further subsections of introduction also discuss the selected frameworks. The methodology used is described in detail and aligns with the research questions. Further, the results and discussion sections are up to the mark and reflect authenticity of this research. However, I would recommend some minor improvements before the publication. The following issues need attention:

• The authors are advised to cite some studies in section 3 “Three-dimensional model of critical discourse” and section 6.2.3 “State-owned vs Private Enterprises” in order to strengthen the arguments.

Afzaal, M. (2023). A corpus-based analysis of discourses on the belt and road 209 initiative: Corpora and the belt and road initiative (Vol. 10). Springer Nature.

Afzaal, M., Hu, K., Chishti, M. I., & Khan, Z. (2019). Examining Pakistan news media discourses about China-Pakistan economic corridor: A corpus-based critical discourse analysis. Cogent Social Sciences, 5(1), 1-18. https://doi.org/10.1080/23311886.2019.1683940

Wei, F. E. N. G., & WU, D. D. (2016). State-owned or Otherwise: Dialogic Construction of Corporate Identities by Chinese Banks on Sina Weibo. Intercultural Communication Studies, 25(2).

Fu, H., & Zhu, H. (2022). Discursive construction of corporate identity through websites: An intercultural perspective on the commercial banks of the United States and China. Frontiers in Psychology, 13, 947012.

• Mention years of the selected frameworks in abstract.

• Avoid the frequent use of contraction (‘s) as in Line 44, 98, 104, 133, 140, 155, 157.

• It is suggested to initiate the paragraphs with articles or cohesive devices as in line 365, 408, 420, 456.

• Use article (the) in lines 451 (the example 25), 486 (the terms), 499, 500,515, 538, 550, 626, 661.

• Use plural form (the examples) in line 485

Re-structure lines 566-567, 716-717 and 583-586.

6. PLOS authors have the option to publish the peer review history of their article (what does this mean? ). If published, this will include your full peer review and any attached files.

**Do you want your identity to be public for this peer review?** For information about this choice, including consent withdrawal, please see our Privacy Policy .

Reviewer #1: No

Reviewer #2: No

---

## [Author Response · Author response to Decision Letter 1]

16 Oct 2024

Dear Dr. Muhammad Afzaal and the PLOS ONE Editorial Team,

Thank you for the opportunity to revise our manuscript entitled "Exploring the Narrative Landscape: The Discursive Construction of Identity for Chinese Enterprises in Africa." We appreciate the constructive feedback from the reviewers and have carefully addressed each point in our revisions.

We have prepared two versions of the manuscript: a marked-up copy highlighting the changes made and an unmarked version for your review. A detailed point-by-point response to the reviewers' comments can be found in the file titled "Response to Reviewers," which also indicates the changes made in the manuscript.

Thank you once again for considering our manuscript for publication in PLOS ONE. We believe our revisions have adequately addressed the reviewers' comments and the journal's requirements, and we are eager to contribute our research to the field.

We look forward to your response.

Sincerely,

Wenyu Liu, PhD

Professor of Linguistics

Dalian University of Technology

No. 2 Linggong Road, Ganjingzi District, Dalian City, Liaoning Province, P.R. China, 116024

liuwy@dlut.edu.cn

+86-139-0408-5717

October 14, 2024

---

## [Editor Report · Decision Letter 1]

8 Nov 2024

Exploring the narrative landscape: the discursive construction of identity for Chinese enterprises in Africa

PONE-D-24-21342R1

Dear Dr. Liu,

We’re pleased to inform you that your manuscript has been judged scientifically suitable for publication and will be formally accepted for publication once it meets all outstanding technical requirements.

Kind regards,

Muhammad Afzaal, PhD

Academic Editor

PLOS ONE
---

## [Editor Report · Acceptance letter]

PONE-D-24-21342R1

PLOS ONE

Dear Dr. Liu,

I'm pleased to inform you that your manuscript has been deemed suitable for publication in PLOS ONE. Congratulations! Your manuscript is now being handed over to our production team.

Kind regards,

on behalf of

Dr. Muhammad Afzaal

Academic Editor

PLOS ONE